# Safety of topical corticosteroids in atopic eczema: an umbrella review

Emma Axon ,[1] Joanne R Chalmers ,[1] Miriam Santer ,[2]
Matthew J Ridd ,[3] Sandra Lawton ,[4] Sinead M Langan ,[5]
Douglas J C Grindlay ,[1] Ingrid Muller ,[2] Amanda Roberts ,[1]
Amina Ahmed ,[1] Hywel C Williams ,[1] Kim S Thomas [1]

► Prepublication history and additional online supplemental material for this paper are available online. To view these files, please visit the journal online (http://dx.doi.org/10.1136/bmjopen-2020-046476).

[1]Centre of Evidence Based Dermatology, University of Nottingham, Nottingham, UK
[2]Primary Care & Population Sciences, University of Southampton, Southampton, UK
[3]Population Health Sciences, University of Bristol Faculty of Health Sciences, Bristol, UK
[4]Dermatology Department, Rotherham NHS Foundation Trust, Rotherham, UK
[5]Faculty of Epidemiology and Population Health, London School of Hygiene & Tropical Medicine, London, UK

**Correspondence to**
Dr Emma Axon;
emma.axon@nottingham.ac.uk

## ABSTRACT

**Objective** An umbrella review summarising all safety data from systematic reviews of topical corticosteroids (TCS) in adults and children with atopic eczema.

**Methods** Embase, MEDLINE, PubMed, Cochrane Database of Systematic Reviews and the Centre of Evidence Based Dermatology map of eczema systematic reviews were searched until 7 November 2018 and Epistemonikos until 2 March 2021. Reviews were included if they assessed the safety of TCS in atopic eczema and searched ≥1 database using a reproducible search strategy. Review quality was assessed using version 2 of 'A MeaSurement Tool to Assess systematic Reviews' (AMSTAR 2 tool).

**Results** 38 systematic reviews included, 34 low/critically low quality. Treatment and follow-up were usually short (2–4 weeks).

**Key findings** TCS versus emollient/vehicle: No meta-analyses identified for skin-thinning. Two 2-week randomised controlled trials (RCTs) found no significant increased risk with very potent TCS (0/196 TCS vs 0/33 vehicle in children and 6/109 TCS vs 2/50 vehicle, age unknown). Biochemical adrenal suppression (cortisol) was 3.8% (95% CI 2.4% to 5.8%) in a meta-analysis of 11 uncontrolled observational studies (any potency TCS, 522 children). Effects reversed when treatment ceased.

*TCS versus topical calcineurin inhibitors:* Meta-analysis showed higher relative risk of skin thinning with TCS (4.86, 95% CI 1.06 to 22.28, n=4128, four RCTs, including one 5-year RCT). Eight cases in 2068 participants, 7 using potent TCS. No evidence of growth suppression.

*Once daily versus more frequent TCS:* No meta-analyses identified. No skin-thinning in one RCT (3 weeks potent TCS, n=94) or biochemical adrenal suppression in two RCTs (up to 2 weeks very potent/moderate TCS, n=129).

*TCS twice/week to prevent flares ('weekend therapy') versus vehicle:* No meta-analyses identified. No evidence of skin thinning in five RCTs. One RCT found biochemical adrenal suppression (2/44 children, potent TCS).

**Conclusions** We found no evidence of harm when TCS were used intermittently 'as required' to treat flares or 'weekend therapy' to prevent flares. However, long-term safety data were limited.

**PROSPERO registration number** CRD42018079409.

## INTRODUCTION

Atopic eczema (also known as atopic dermatitis or eczema) is an itchy inflammatory skin condition. It is most common in children

## Strengths and limitations of this study

► Robust Cochrane methodology was followed and a thorough and inclusive literature search was performed to ensure this was a comprehensive overview.

► By extracting data from existing reviews, results are limited to topics for which there is an eligible systematic review.

► Safety was usually reported in less detail than effectiveness in systematic reviews limiting the available data for extraction, therefore potentially missing data included in the original papers.

► Most included reviews were rated low or critically low-quality using AMSTAR 2, and where quality of evidence assessments were reported for individual studies most indicated a high or unclear risk in at least one domain.

► Many randomised controlled trials were only short in duration (2–4 weeks) limiting our ability to assess side effects that take longer to develop such as skin thinning.

with one in five affected worldwide,[1 2] but often persists into adulthood.[3]

Topical corticosteroids (TCSs) are first-line therapy for treating inflammatory eczema flares but widespread concerns regarding their safety among patients and healthcare professionals contribute to poor adherence, and subsequent worsening of disease control and quality of life.[4 5] Safety concerns include skin thinning and retardation of growth and development. These concerns are thought to mainly originate from what is now considered to be inappropriate use, such as using potent TCS on the face or continual long-term use.[6] Strategies recommended to minimise exposure to TCS, and hence the risk of adverse events, include reducing frequency of application to once daily during treatment of an inflammatory episode, or TCS used for two consecutive days a week (sometimes referred to as 'weekend therapy') as a strategy to prevent flares.[7–9] This umbrella review aims

to assess safety (local and systemic adverse events) of TCS compared with other topical treatments, placebo or no comparator in people of any age and gender with atopic eczema, and addressed two areas of research prioritised in the James Lind Alliance priority setting partnership for atopic eczema.[10]

## METHODS

### Protocol, registration and study design

This umbrella review includes published systematic reviews of randomised controlled trials (RCTs) and/or observational studies reporting adverse event data in people with eczema using TCS. The aim of this overview was to summarise data from existing reviews, therefore, meta-analyses and data from individual studies were extracted and presented in this overview in the format and completeness that they were presented in the original systematic reviews. The only exception was for missing p values which were calculated where appropriate. The checklist 'Preferred Reporting Items for Systematic Reviews and Meta-Analyses (PRISMA)' was followed.[11 12]

### Search strategy

Embase, MEDLINE, PubMed, Cochrane Database of Systematic Reviews and Epistemonikos were searched from inception to 7 November 2018 by DJCG (information specialist), with no restrictions on language or publication date. The search strategy is in online supplemental appendix 1. The Epistemonikos search was updated on 2 March 2021, with a publication date restricted to 2018–2021. Epistemonikos is now well established as a comprehensive database of reviews that regularly searches ten major databases including the Cochrane Library, PubMed and Embase[13] thus making the need to search these individual databases redundant. We also checked the Centre of Evidence Based Dermatology eczema map of systematic reviews,[14] and searched PROSPERO up to 23 March 2021 for any relevant ongoing systematic reviews using the terms 'eczema' and 'dermatitis'.

### Eligibility criteria

We included systematic reviews that presented data on the safety of TCS used to treat people of any age and gender with atopic eczema, had clinical outcomes, searched at least one database and provided a reproducible search strategy. Systematic reviews of any types of clinical study design were included. Multiple reviews on the same topic were included, except for 'abridged' versions of the same review where no additional data were reported. To avoid duplication of data, for each comparison, the review that included the highest number of studies on that comparison and therefore appeared the most comprehensive was taken as the primary review and other included reviews were checked for additional studies and data. Conference abstracts were excluded. Reviews that covered multiple skin conditions were only included if they reported data on atopic eczema patients separately.

### Interventions and control

Our intervention of interest was any TCS of any preparation and potency used to treat atopic eczema. For RCTs, the comparisons of interest were any other TCS, the same TCS used in a different way, another topical anti-inflammatory treatment, vehicle, no treatment or a combination of any of these. Comparisons with non-topical treatments were excluded as we were interested in clinical practice decisions regarding alternatives to TCS.

### Outcomes

Safety outcomes reported during the treatment and follow-up were extracted where reported in the reviews on immediate cutaneous adverse events (eg, burning sensation/stinging), other cutaneous adverse events (eg, skin thinning, telangiectasia, skin infections, folliculitis), systemic adverse events (eg, effects on endocrine system, impact on growth) and rebound symptoms/steroid withdrawal.

### Selection of studies and data extraction

Records identified from the database searches were uploaded into Covidence (Veritas Health Innovation, Australia).[15] Two authors (EA and JRC) independently assessed the eligibility of each record, and where unclear the full text was obtained. The number of included and excluded records along with reasons for exclusion were reported in a PRISMA flow diagram.

Two authors (EA and JRC) independently extracted all safety data presented in the included reviews along with other information such as review/participant characteristics, and funding sources. Any disagreements regarding eligibility or data extraction were resolved via discussion or input from a third reviewer (HCW or KST). Where available, we reported results separately for age, filaggrin mutation status, TCS potency, site of application of the TCS, and duration of continuous treatment.

### Assessment of quality of included systematic reviews

As this was an overview of reviews, the methodological quality of the evidence was assessed at the systematic review level using version 2 of 'A MeaSurement Tool to Assess systematic Reviews' (AMSTAR 2 tool) and this was conducted in duplicate by EA and JRC.[16] Reviews were considered critically low quality if there was more than one critical flaw. Data on the quality of individual studies (eg, risk of bias) and the quality of evidence (eg, Grading of Recommendations Assessment, Development and Evaluation, GRADE[17]) were also extracted where presented in the review, but undertaking these quality assessments for individual studies was not within the remit of this overview.

### Measures of treatment effect and data synthesis

Where relevant meta-analyses were presented in the systematic review, the forest plots, relative risk (RR) and 95% CI were extracted. In the absence of any meta-analysis, adverse event data from individual studies were included in this overview based on the data presented in

the published systematic review. P values were calculated using Review Manager software,[18] with <0.05 indicating statistically significant results. Where meta-analyses were presented, we assessed the following subgroups where possible: age, TCS potency, anatomical site, treatment duration and genetic predisposition to a disrupted skin barrier (filaggrin status). TCS potency was determined using a hierarchy of sources: UK 'British National Formulary', WHO and USA classifications.[19–21] A National Health Service classification ranging from very common (>1 in 10 people affected) to very rare (<1 in 10 000) was used to narratively describe the absolute risk of each adverse event.[22]

## Patient and public involvement

People with eczema and parents of children with eczema were involved in the decision to conduct this overview and in the design. The James Lind Alliance priority setting partnership for atopic eczema involved people with eczema and parents of children with eczema in which two of the identified priority areas were around research into the safety of TCS.[10] Two of the overview authors are patient representatives (AR and AA) and both have been involved in the design of this overview and interpretation of the findings.

Wider patient and parent involvement has been particularly important in identifying important safety outcomes for this overview. We held a workshop involving five patient representatives in which the proposed overview was discussed which highlighted the need to seek out data on long-term TCS use, reversibility of any side effects and TCS withdrawal symptoms. We supplemented this with a survey about safety concerns with TCS at a National Eczema Society meeting of 31 people with eczema or parents of children with eczema and a published qualitative study of patient concerns relating to TCS safety.[6]

Dissemination of the results is underway as part of the wider programme of research of which this overview is a part and our patient representatives are a key stakeholder in this activity.

## RESULTS
## Search results

After deduplication, 635 records were screened; 127 records underwent full-text screening and 38 systematic reviews met the inclusion criteria (figure 1).[7 8 23–56] The list of excluded reviews is in online supplemental appendix 2. The search of PROSPERO identified five ongoing systematic reviews (online supplemental appendix 3).[57–61]

## Characteristics and quality of the included systematic reviews

All but three reviews were published in English. Two Chinese reviews and one German review were translated into English.[32 36 45] Thirty of the included reviews were rated critically low quality according to AMSTAR 2; with four low, two moderate and two high quality (table 1). The most common reasons for downgrading were no

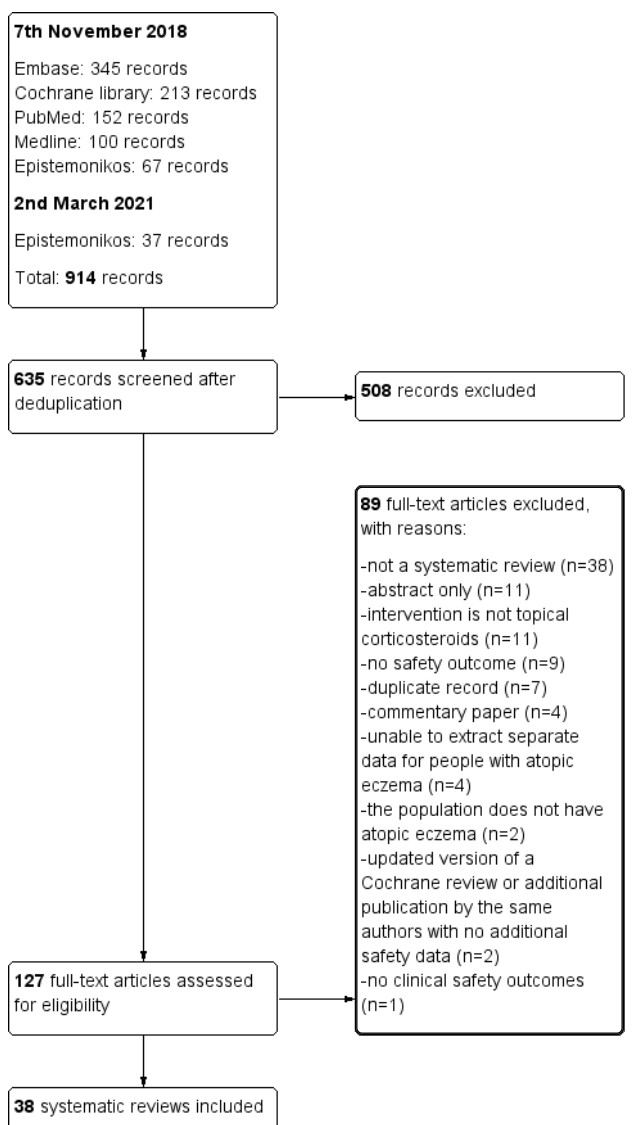

**Figure 1** PRISMA flow diagram. PRISMA, Preferred Reporting Items for Systematic Reviews and Meta-Analyses.

protocol, no list of full-text exclusions or a literature search restricted to the English language.

The included reviews identified 106 studies (77 RCTs and 29 observational studies) that included relevant safety data. Risk of bias assessments were available from the reviews for 63 RCTs, of which 42 used the Cochrane risk of bias tool. Most of these assessments rated at least one domain as high or unclear risk, most noticeably selection bias from lack of allocation concealment, performance bias due to lack of blinding of participants and detection bias due to lack of blinding of outcome assessors. The trials included in the reviews usually evaluated use of short bursts of TCS (2–4 weeks) to treat the flare but varied greatly in length of follow-up. Around two-thirds of trials included no post-treatment follow-up, while the remainder included several weeks/months of follow-up generally using TCS intermittently 'as required'. A total of 14 RCTs (5874 participants) and 5 cohort/observational

**Table 1** Characteristics of included systematic reviews

| First author, publication year | Type of review | Review contained safety data from RCTs for comparisons of interest? | Review contained safety data from observational studies? | AMSTAR 2 rating |
|---|---|---|---|---|
| Ashcroft 2005[24] | Non-Cochrane | Yes (TCS vs TCI) | No | Critically low [1 3 6 7] |
| Ashcroft 2007[23] | Cochrane | Yes (TCS vs TCI) | Yes (TCS vs TCI) | Moderate[8] |
| Barnes 2015[25] | Non-Cochrane | Yes (TCS vs vehicle, TCS vs TCI, TCS vs another TCS) | Yes (single arm TCS studies) | Critically low [1 2 3 4 6] |
| Braham 2010[26] | Non-Cochrane | Yes (occluded TCS vs non-occluded TCS) | Yes (occluded TCS) | Critically low [1 2 3 4 6] |
| Broeders 2016[27] | Non-Cochrane | Yes (TCS vs TCI) | No | Critically low [1 3 5 6] |
| Callen 2007[28] | Non-Cochrane | Yes (TCS vs vehicle, TCS vs another TCS) | Yes (single arm studies or comparing TCS potencies) | Critically low [1 2 3 4 6] |
| Chen 2010[29] | Non-Cochrane | Yes (TCS vs TCI) | No | Critically low [1 3 6] |
| Cury Martins 2015[30] | Cochrane | Yes (TCS vs TCI) | Yes (TCS vs TCI) | Moderate[8] |
| De Tiedra 1997[31] | Non-Cochrane | Yes (TCS vs another TCS) | Yes (usually only reported data from one arm of RCTs) | Critically low [1 2 3 4 6] |
| Devillers 2006[32] | Non-Cochrane | Yes (occluded TCS vs non-occluded TCS) | Yes (occluded TCS) | Critically low [1 2 3 4 6] |
| Dong 2017[33] | Non-Cochrane | Yes (TCS vs TCI) | No | Critically low [1 2 3 4 6] |
| Eichenfield 2014[34] | Non-Cochrane | No | Yes (different TCS potencies) | Critically low [1 2 3 4 6] |
| Feldman 2005[35] | Non-Cochrane | Yes (TCS vs vehicle) | No | Critically low [1 2 3 4 6] |
| Fishbein 2019[63] | Non-Cochrane | Yes (TCS vs vehicle/moisturiser) | No | Critically low [3 4 5 6 7] |
| Frangos 2008[36] | Non-Cochrane | Yes (TCS vs vehicle) | Yes (single arm studies) | Critically low [1 2 3 4 6] |
| Froeschl 2007[37] | GMS HTA report | Yes (TCS vs vehicle, TCS vs TCI, TCS vs another TCS) | No | Critically low [1 2 4 6] |
| Gonzalez-Lopez 2017[38] | Non-Cochrane | Yes (occluded TCS vs non-occluded TCS) | No | Critically low [1 3] |
| Green 2004[7] | HTA report | Yes (once daily vs twice daily TCS use) | No | Low |
| Gu 2013[40] | Cochrane | Yes (TCS vs topical CHM) | No | High |
| Gu 2014[39] | Non-Cochrane | Yes (TCS vs topical CHM) | No | Critically low [1 2 3 7] |
| Hajar 2015[41] | Non-Cochrane | No | Yes (case series or case reports) | Critically low [2 3 6] |
| Hoare 2000[42] | NIHR HTA report | Yes (TCS vs vehicle, TCS vs another TCS) | No | Low |
| Iskedjian 2004[43] | Non-Cochrane | Yes (TCS vs vehicle, TCS vs TCI) | No | Critically low [1 3 6] |
| Juhász 2017[44] | Non-Cochrane | No | Yes (social media analysis) | Critically low [1 2 3 4 6] |
| Abędź 2019[82] | Non-Cochrane | Yes (TCS vs TCI) | No | Critically low [1 3 6 7] |
| Legendre 2015[45] | Non-Cochrane | No | Yes (TCS vs TCI) | Critically low [1 2 3 6] |
| Li 2007[46] | Non-Cochrane | Yes (TCS vs TCI) | No | Critically low [1 3 6] |
| Nankervis 2016[47] | NIHR HTA report | Yes (TCS vs vehicle, TCS vs emollients, TCS vs TCI, TCS vs another TCS, once a day vs twice a day use, proactive TCS to prevent flares ('weekend therapy') vs vehicle, occluded TCS vs non-occluded TCS) | No | Low |

**Table 1** Continued

| First author, publication year | Type of review | Review contained safety data from RCTs for comparisons of interest? | Review contained safety data from observational studies? | AMSTAR 2 rating |
|---|---|---|---|---|
| Burls 2004[48] | West Midlands HTA report | Yes (TCS vs TCI) | No | Low |
| Schmitt 2011[8] | Non-Cochrane | Yes (proactive TCS to prevent flares ('weekend therapy') vs vehicle) | No | Critically low [3 6] |
| Sidbury 2014[49] | Non-Cochrane | Yes (proactive TCS to prevent flares ('weekend therapy') vs vehicle) | No | Critically low [1 2 3 4 6] |
| Siegfried 2016[50] | Non-Cochrane | Yes (TCS vs vehicle, TCS vs TCI, TCS vs another TCS) | No | Critically low [1 2 3 4 6] |
| Singh 2012[51] | Non-Cochrane | Yes (TCS vs vehicle, TCS vs TCI, TCS vs another TCS) | Yes (single arm study) | Critically low [1 2 6] |
| Svensson 2011[52] | Non-Cochrane | Yes (TCS vs TCI) | No | Critically low [1 3 6 7] |
| Tang 2014[53] | Non-Cochrane | Yes (proactive TCS to prevent flares ('weekend therapy') vs vehicle) | No | Critically low [1 3 4 6] |
| van Zuuren 2017[54] | Cochrane | Yes (TCS vs emollient) | No | High |
| Wood Heickman 2018[55] | Non-Cochrane | No | Yes (single arm cohort studies) | Critically low [1 2 3 4 6 7] |
| Yan 2008[56] | Non-Cochrane | Yes (TCS vs TCI) | No | Critically low [1 3 6 7] |

AMSTAR 2 ratings—reasons for downgrading the quality of the review: [1]No protocol; [2]Search strategy not comprehensive; [3]No list of excluded studies with reasons; [4]Risk of bias not assessed; [5]Inappropriate meta-analysis methods; [6]Risk of bias assessments not included in the interpretation of the results; [7]Publication bias not explored in the meta-analysis.
Additional data on TCS including potency can be found in online supplemental appendix 6.
CHM, Chinese herbal medicine; GMS, German Medical Science; HTA, Health Technology Assessment; NIHR, National Institute for Health Research; RCTs, randomised controlled trials; TCI, topical calcineurin inhibitors; TCS, topical corticosteroid.

studies (4 438 698 participants) out of a total of 106 studies included follow-up of more than 3 months. One notable trial (the 'PETITE' study) had 5 years follow-up with TCS used 'as required'.[62]

Characteristics and quality assessments of each systematic review are in table 1, with further detail in online supplemental appendices 4 and 5. Individual study data and quality assessments are in online supplemental appendix 6.

### Safety of TCS compared with other topical treatments or corticosteroids

#### How safe are TCS compared with emollient or vehicle, or no comparison?

Thirteen reviews provided data on this comparison: 1 high[54], 2 low[42 47] and 10 critically low quality.[25 28 31 35–37 50 51 55 63] Key results can be found in table 2 and additional data in online supplemental appendix 6.

Reported rates of skin thinning in RCTs were generally very low, with no significant increases seen with TCS compared with emollient/vehicle. No skin thinning or telangiectasia was reported in an RCT, 196 participants aged ≥12 years old using very potent TCS twice a day for 2 weeks compared with 33 using vehicle.[64] Another RCT reported skin thinning in 6/109 participants using

very potent TCS for 2 weeks compared with 2/50 using vehicle, p=0.69.[65]

No significant differences in other cutaneous adverse events, such as hypopigmentation, were observed between treatments in five RCTs, and event rates were low.[66–70]

A meta-analysis[55] of 11 uncontrolled observational studies (up to 4 weeks of treatment) reported biochemical adrenal suppression (cortisol levels) in 20/522 children (3.8%, 95% CI 2.4% to 5.8%) with any potency TCS.[71–81] This was 2% (3/148 children) when only mild potency TCS were analysed.[72 74 77 79] No clinical symptoms or signs of adrenal suppression were observed,[71–81] and the biochemical effects were transient, with cortisol levels returning to normal after TCS were discontinued.[71 75 77 78 81]

Two included reviews assessed TCS withdrawal symptoms, mostly from case reports, but no incidence data were reported.[41 44]

#### How safe are TCS compared with topical calcineurin inhibitors?

Eight systematic reviews were identified: one moderate[23], one low[48] and six critically low quality.[27 30 43 50 52 82] Most RCTs used twice daily TCS to treat the current flare (up to 3 weeks), and where longer-term follow-up was included, TCSs were used 'as required' to treat flares. Key results

**Table 2** Summary of main findings for key safety outcomes

| | Cutaneous adverse events | Systemic adverse events |
|---|---|---|
| **How safe are TCS compared with emollient or vehicle, or no comparison?** 13 reviews: 1 moderate quality 2 low quality 10 critically low quality | ▶ **Skin thinning:** No significant differences in 2 RCTs of 2–4 weeks compared with emollient/vehicle: (1) 0/196 children with very potent TCS and 0/33 vehicle, (2) 6/109 very potent TCS vs 2/50 vehicle, p=0.69. Very low rates. ▶ **Other cutaneous adverse events:** No significant differences in 5 RCTs (2–4 weeks) between TCS (various potencies) and emollient/vehicle (n=172, plus one study, n not specified). Low event rates. | ▶ **Biochemical evidence of adrenal suppression:** Meta-analysis (11 observational studies, max 4 weeks)—20/522 children with any potency TCS (3.8%, 95% CI 2.4% to 5.8%), 3/148 children (2%) with mild potency TCS. Effects were transient. ▶ **Clinical symptoms or signs of adrenal suppression:** none observed in same as above observational studies. |
| **How safe are TCS compared with topical calcineurin inhibitors (TCI)?** 8 reviews: 1 moderate quality 1 low quality 6 critically low quality | ▶ **Skin thinning:** Higher with TCS than TCI (meta-analysis of 4 RCTs: RR 4.86, 95% 1.06 to 22.28, n=4128) but very low rate (8/2068, 7 of which were using potent TCS). ▶ **Other cutaneous adverse events:** No difference in **skin infections** between TCS and TCI (8 RCTs). **Skin burning** and **pruritus** lower with TCS than TCI: meta-analysis of 10 RCTs: **burning**—RR 0.31, 95% CI 0.23 to 0.40 (n=4211), **pruritus**—RR 0.68, 95% CI 0.56 to 0.82(n=4211). | ▶ **Growth rate:** no differences in growth rates tween TCS and TCI (1 RCT of 2418 children with 5 years follow-up). ▶ **Lymphoma:** no cases reported in one same large RCT as above. One cohort study (n=1 438 333, approx. 4 years follow-up)—very small non-significant increase with TCI *and* TCS compared with general population. One case–control study—no increased risk with TCS or TCIs (294 cases/293 000 controls). |
| **How safe are once daily TCS compared with twice daily application?** 2 reviews: 2 low quality | ▶ **Skin thinning:** no cases using once daily vs twice daily potent TCS for 3 weeks (1 RCT, 94 adults). ▶ **Other cutaneous adverse events:** no significant difference between groups in **telangiectasia**, **folliculitis**, or **burning/itching/stinging** (4 RCTs, 4–16 weeks follow-up 740 older children/adults). | ▶ **Biochemical evidence of adrenal suppression:** no significant differences between once and twice daily moderate/potent TCS up to 2 weeks in children (2 RCTs, n=129). |
| **How safe are TCS used proactively to prevent flares ('weekend therapy')?** 3 reviews: 3 critically low quality | ▶ **Skin thinning:** no cases with 16–20 weeks of 2 days/week of potent TCS vs vehicle (5 RCTs, n=993). ▶ **Other cutaneous adverse events:** no significant differences between groups, including **folliculitis** and transient **telangiectasia**, with potent TCS (16–20 weeks) compared with either vehicle or another TCS (2 RCTs, n=423). Events were uncommon in both groups. | ▶ **Biochemical evidence of adrenal suppression:** no cases with 16 weeks of 2 days/week of potent TCS (2 RCTs, n=129). Possible adrenal suppression in 2/44 children with potent TCS compared with zero using vehicle (1 RCT, 20 weeks). |
| **How safe are TCS used under occlusion?** 4 reviews: 1 high quality 3 critically low quality | ▶ **Skin thinning:** no cases in two observational studies (potent TCS +wet wrap, 1–2 weeks, n=44). ▶ **Other cutaneous adverse events:** One case of **striae** in two observational studies, n-44. More **folliculitis** with diluted potent TCS (10/19 children) compared with emollient (2/20), both under wet wrap (1 RCT). A meta-analysis (2 RCTs, n=69) of wet wrap vs no wet wrap (mild potency)—no significant difference in **cutaneous adverse events**. | ▶ **Biochemical evidence of adrenal suppression:** reported in three *observational* studies (2–14 days of diluted potent TCS under wet-wraps in 74 children) but rates not specified in review. Described as transient in two studies. ▶ **Growth or bone turnover:** no effect seen in one small short-term *observational* study (potent TCS wet-wrap in eight children, (median follow-up 12 weeks). |

RCTs, randomised controlled trials; RR, relative risk; TCS, topical corticosteroids.

can be found in table 2 and additional data in online supplemental appendix 6.

Meta-analyses of cutaneous adverse events were presented in two reviews.[27 82] So the more comprehensive review was used to extract the cutaneous adverse event data.[27] Some minor modifications were made to the data for this overview shown in online supplemental appendix 7. A meta-analysis of four RCTs (26 weeks to 5 years duration, twice a day or 'as directed') showed a significant increase in the RR of skin thinning with TCS compared with topical calcineurin inhibitors (TCIs) (0.1% tacrolimus or 1% pimecrolimus) (RR 4.86, 95% CI 1.06 to 22.28, p=0.04, n=4128). However, skin thinning was uncommon: 8/2068 participants (0.4%) with TCS vs 0/2060 (0%) with TCIs. Of the eight cases of skin thinning, seven were reported when using potent TCS and one using mild/moderate TCS.[62 83–85]

The RR of skin burning and pruritus (itching) was significantly lower with TCS compared with TCIs (1% pimecrolimus or 0.1 % / 0.03% tacrolimus) in meta-analyses of 10 RCTs in 4211 participants (skin burning: RR 0.31, 95% CI 0.23 to 0.40, p<0.00001; pruritus: RR 0.68, 95% CI 0.56 to 0.82, p<0.0001).[83 85–93] The GRADE assessments for these two adverse events indicated these were of moderate quality.[82] There was no significant difference in skin infections with potent, moderate or mild potency TCS compared with TCIs (1% pimecrolimus or 0.1 %/0.03% tacrolimus)[62 83–86 88 90 92] or erythema compared with 0.1% tacrolimus (online supplemental appendix 8).[91 92]

Subgroup analyses of age, TCS potency and specific TCI showed no significant differences for any comparison (online supplemental appendix 9). We were unable to undertake any further subgroup analyses.

No differences in growth were observed in one 5-year RCT ('PETITE' study) in 2418 young children using moderate/mild potency TCS compared with those using TCI (1% pimecrolimus) (rates not given) and no cases of lymphoma were reported.[62] A large cohort study (n=1 438 333) showed a small non-significant increased risk of lymphoma with TCI and TCS compared with the general population, with a similar risk between treatments.[94] In addition, one case–control study (294 cases/293 000 controls) found no increased risk of lymphoma with TCS or TCI compared with controls.[95]

### Is there any difference in safety of TCS of different potencies?

Six reviews compared the safety of different potency TCS: two low,[42 47] and four critically low quality.[28 34 50 53] RCTs were mainly short-term use of TCS (2–3 weeks), used once or twice daily. Results can be found in online supplemental appendix 6.

One RCT reported mild skin thinning in 4/13 children using potent TCS for up to 6 weeks compared with 2/12 using mild TCS (p=0.42),[96] while another RCT in 37 children found no evidence of skin thinning with mild or moderate potency TCS for 3 weeks.[97] One study compared 3 weeks of potent and moderate TCS in

40 children and reported 'some' biochemical adrenal suppression (cortisol levels) but no numerical data were provided.[98]

### How safe are TCS compared with topically applied Chinese herbal medicine?

Two systematic reviews provided data on TCS compared with topical Chinese herbal medicine: one high quality[40] and one critically low.[39] Results can be found in online supplemental appendix 6.

A meta-analysis of two RCTs[99 100] was presented in two systematic reviews.[39 40] More cutaneous adverse events, including application site burning, were observed with 2 weeks of very potent/potent TCS compared with topical Chinese herbal medicine (RR 12.03, 95% CI 1.59 to 91.26, p=0.02; 11/147 vs 0/148 participants). One additional RCT, including 95 young children, reported minor adverse events such as burning with 2 weeks of potent TCS but no numerical data were presented.[101]

## Safety of different strategies for using TCS

### How safe are once daily TCS compared with more frequent application?

Two low-quality reviews provided safety data relating to different frequency of application.[7 47] Key results can be found in table 2 and additional data in online supplemental appendix 6.

No skin thinning was reported with once or twice daily application of potent TCS for 3 weeks in one RCT (94 adults).[102] Four RCTs in 740 older children/adults showed no significant difference between once and twice daily application of moderate/potent TCS in other cutaneous adverse events including telangiectasia,[103 104] folliculitis[105] and burning, itching or stinging.[105 106] Two RCTs showed no significant differences in biochemical adrenal suppression (cortisol levels) between once and twice daily very potent/moderate TCS used for up to 2 weeks in 129 children.[81 107]

### How safe are TCS when used proactively to prevent flares ('weekend therapy')?

Two reviews included data on the safety of TCS used proactively 2 days a week ('weekend therapy') to prevent flares, both critically low quality.[8 53] Key results can be found in table 2 and additional data in online supplemental appendix 6.

There was no evidence of skin thinning in five RCTs comparing 16–20 weeks of weekend therapy with potent TCS versus vehicle in 993 participants.[103 108–111] Furthermore, two RCTs (n=423) reported no significant differences in other cutaneous adverse events, including folliculitis and transient telangiectasia, with potent TCS compared with vehicle.[108 109] Events were uncommon in both groups.

There was no evidence of biochemical adrenal suppression (cortisol levels) in two RCTs (n=129) between potent TCS and vehicle used for 16 weeks.[108 111] In a 20-week

RCT, 2/44 children had possible adrenal suppression with potent TCS compared with zero with vehicle.[109]

### How safe are TCS used under occlusion?

Four reviews included data on the safety of TCS used under occlusion: one high[54], and three critically low quality.[26 32 38] Results can be found in online supplemental appendix 6.

There were no cases of skin thinning and one case of striae in two uncontrolled observational studies of a diluted potent TCS used under wet-wrap for 1–2 weeks in 44 young children.[112 113] A significant difference in the rate of folliculitis (mostly mild) was observed in one RCT of TCS under wet-wrap for 4 weeks, with more folliculitis in the diluted potent TCS group (10/19 children) compared with emollient (2/20 children) (p=0.02).[114] A meta-analysis from one review[38] of two RCTs in young children showed no significant difference in the number of participants with cutaneous adverse events between mild potency TCS under wet wrap (7/38 participants) versus not under wet-wrap (0/31 participants) (p=0.08)[115 116]; this evidence was rated low quality by the systematic review authors using GRADE.[17]

Biochemical adrenal suppression (cortisol levels) was reported in three uncontrolled observational studies of 2–14 days of diluted potent TCS under wet-wraps in 74 children.[112 113 117] Actual rates were not specified in the review, but increases were described as transient in two studies.[112 117] One short-term uncontrolled observational study of diluted potent TCS under wet-wrap in eight children showed no effect on growth or bone turnover.[118]

### DISCUSSION

This comprehensive overview of systematic reviews which, for the first time, brings together all safety data from systematic reviews on TCS used in eczema from 38 systematic reviews, a topic that was identified as a priority in a James Lind Alliance priority setting partnership on eczema. Skin thinning and effects on growth concern many people with eczema and parents of children with eczema when using TCS. However, we found no evidence of skin thinning when TCS were used intermittently 'as required' to treat flares or as 'weekend therapy' to prevent flares, although the majority of data was from short-term studies.[5] Similarly, we found no evidence of growth retardation or clinically significant adrenal suppression but the only data available was from one 5-year study that included 1213 children using TCS.[62] Other studies only reported biochemical signs of adrenal suppression. Adherence to TCS treatment is known to be poor and these findings, particularly around skin thinning, may encourage appropriate use of TCS and therefore improve treatment effectiveness and patient benefit.[119]

A thorough literature search was conducted and Cochrane methodology was used. Conclusions were limited by the content of the included reviews because safety was frequently reported in less detail than effectiveness, reviews reported on different adverse events and some adverse events were not described in the reviews. It is not clear whether this is because the trials did not report adverse events in sufficient detail or whether the review authors did not include all the available safety data, perhaps only focusing on a restricted group of adverse events. None of the included systematic reviews presented data on our prespecified subgroup analyses. Furthermore, most of the included reviews were rated low or critically low-quality using AMSTAR 2. The lack of comprehensive search strategies and duplicate screening/data extraction in the included reviews may have resulted in missing studies and safety data, which could have impacted on this overview particularly where there was limited data. In addition, where the quality of evidence assessments (eg, GRADE) were reported in the reviews, most individual studies included in the reviews indicated a high or unclear risk in at least one domain.

Many RCTs did not include follow-up beyond 2–4 weeks of treatment and therefore data on long-term safety are limited. Although short-term TCS use reflects appropriate treatment duration for treating an individual flare, it does not reflect the chronic nature of eczema and the need for TCS use over the long-term. The 'PETITE study' was the notable exception and data published in the correspondence showed there was only one episode of skin thinning in 1213 children using mild/moderate TCS 'as required' with 5-year follow-up.[62] Trials using intermittent TCS as 'weekend therapy' to prevent flares also provide reassurance for the safety of longer-term use of TCS, as these trials generally included 16–20 weeks of follow-up to assess the prevention of flares. The inclusion of systematic reviews that included observational studies as well as reviews of RCTs also increased the amount of safety data available to report in this overview.

Although this review focused on the safety of TCS as the key issue for patients, treatment decisions are a balance of benefits and harms. For example, although the safety profile of Chinese herbal medicine was better than TCS, in practice this would be considered alongside the relative effectiveness of these treatments. Likewise, although there was no difference in the safety of once vs twice daily TCS, effectiveness of these regimens is also important to consider. A Cochrane review is underway comparing the effectiveness and safety of different ways of using TCS.[120]

In summary, we found no evidence that TCS cause harm when used intermittently 'as required' to treatment eczema flares or as 'weekend therapy' to prevent flares and this should support the use of TCS in the management of eczema. We found that the adverse events of greatest concern to patients and clinicians, such as skin thinning, are uncommon with short-term use of TCS. However, high-quality evidence was limited, particularly for long-term use. Rather than follow-up of perhaps just a few weeks, future RCTs should include lengthier follow-up to enable better safety assessment. However, it should be noted that longer-term prospect observational studies are better placed to explore longer-term safety of TCS and

should be designed with years rather than months of follow-up to add useful information to the field. Perhaps equally as important as duration of follow-up in trials is resolution of adverse events which is often not reported. For adverse events such as biochemical signs of adrenal suppression, it is crucial to know if the effect is transient and levels return to normal once the TCS is stopped, particularly as it is not clear how to interpret the clinical relevance of these.

**Acknowledgements** We would like to thank Faye Shelton for her assistance with the screening of the search results and to Jane Harvey for her input into classifying the TCS potencies. We would also like to thank Chiau Ming Long for translating two of the included reviews published in Chinese, and to Jonathan Batchelor for confirming exclusion of two reviews published in Japanese.

**Contributors** All authors (EA, JRC, MS, MJR, SL, SML, DJCG, IM, AR, AA, HCW and KST) helped conceive of and design this overview. DJCG and EA conducted the searches. EA and JRC carried out the eligibility screening, data extraction and quality assessments. HCW and KST acted as 3rd reviewers to resolve disagreements. EA performed the statistical analysis and JRC is the study guarantor. EA and JRC collated and interpreted the data with input from all other authors. EA and JRC completed the initial drafts of the manuscript and all authors (EA, JRC, MS, MJR, SL, SML, DJCG, IM, AR, AA, HCW and KST) commented on and approved the final manuscript. The corresponding author attests that all listed authors meet authorship criteria and that no others meeting the criteria have been omitted.

**Funding** This report presents independent research funded by the National Institute for Health Research (NIHR) under its Programme Grants for Applied Research programme (grant ref No. RP-PG-0216-20007).

**Disclaimer** The views expressed are those of the author(s) and not necessarily those of the NHS, the NIHR or the department of Health and Social Care.

**Competing interests** Authors are coapplicants on an NIHR Programme Grants for Applied Research (P-PG-0216-20007) which funded this overview. The aim of the Programme Grant is to develop an intervention to support eczema self-care and the results of this overview will contribute to this intervention. MJR is funded by an NIHR Post-Doctoral Research Fellowship (PDF-2014-07-013). SML is supported by a Wellcome Senior Clinical fellowship in Science (205039/Z/16/Z). HCW was an author on four included reviews, and KST was an author on one included review.

**Patient consent for publication** Not required.

**Provenance and peer review** Not commissioned; externally peer reviewed.

**Data availability statement** Data are available on reasonable request. All data relevant to the study are included in the article or uploaded as online supplemental information. For any further details email cebd@nottingham.ac.uk or emma.axon@nottingham.ac.uk

**ORCID iDs**
Emma Axon http://orcid.org/0000-0002-3246-9968
Joanne R Chalmers http://orcid.org/0000-0002-2281-7367
Miriam Santer http://orcid.org/0000-0001-7264-5260
Matthew J Ridd http://orcid.org/0000-0002-7954-8823
Sandra Lawton http://orcid.org/0000-0002-6163-5822
Sinead M Langan http://orcid.org/0000-0002-7022-7441
Douglas J C Grindlay http://orcid.org/0000-0002-0992-7182
Ingrid Muller http://orcid.org/0000-0001-9341-6133
Amanda Roberts http://orcid.org/0000-0003-0370-3695
Amina Ahmed http://orcid.org/0000-0001-9494-742X
Hywel C Williams http://orcid.org/0000-0002-5646-3093
Kim S Thomas http://orcid.org/0000-0001-7785-7465

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
