## [Reviewer comments · BMJ Open]

ARTICLE DETAILS

TITLE (PROVISIONAL)	Safety of topical corticosteroids in atopic eczema; an umbrella review
AUTHORS	Axon, Emma; Chalmers, Joanne; Santer, Miriam; Ridd, Matthew; Lawton, Sandra; Langan, Sinead; Grindlay, Douglas; Muller, Ingrid; Roberts, Amanda; Ahmed, Amina; Williams, HC; Thomas, Kim

VERSION 1 – REVIEW

REVIEWER	Yiu, Zenas Salford Royal NHS Foundation Trust
REVIEW RETURNED	22-Dec-2020

GENERAL COMMENTS	This is a well-reported well-conducted umbrella review for the safety of topical corticosteroids in the treatment of atopic eczema, which is a very important topic for both patients and clinicians. Comments: Consider updating the search as it will have been more than 2 years since the search date by the time of publication. Consider adding whether authors on this review were also authors in the SRs under consideration in your competing interest section. It may be worth adding the detail over participants and study design in your PICOS study question definition in the introduction. The section on comparator/control from the protocol doesn't seem to have been included in the main manuscript. It would be really useful if you could summarise exactly how many studies, and participants, had longer term, say >3/12, data available for safety. Further to this point, it may be worth presenting or framing AEs as short term AEs vs long term AEs. Given the majority of the evidence is in the short-term, the findings could be more reassuring for short-term AEs, say folliculitis, infections, and not so informative for skin thinning, telangiectasia and systemic AEs. How was a "systematic" review defined? (in protocol but not in manuscript – I think it should be included.)
--

	How were skin thinning and steroid withdrawal defined in the studies and in the SRs? Did the authors consider formulation, e.g. foam vs ointment vs cream, for subgroup analysis? It could be relevant for short-term AEs, e.g. sensations of skin burning. If not, is this a limitation to consider? There are reference errors on line 56, page 8; line 39, page 9; line 54, page 10. What were the methods used for quality assessment? Were they also done in duplicate? Any assessment of meta-biases across studies? A presentation or discussion of the overlapping of studies between SRs may be useful. I would suggest maybe caveating the first paragraph with your finding of lack of longer-term data, as you have done in the conclusion, as I am not sure this is really reliable data to reassure patients and aid adherence given the paucity of longer-term data and low quality of SRs. How long should RCTs include for follow-up at a minimum in your suggestion in the conclusion? Worth being specific here. How about cohort studies? I suggest adding "in the short-term" to your last sentence.
--	--

REVIEWER	Gooderham, Melinda Queen's University
REVIEW RETURNED	20-Jan-2021

GENERAL COMMENTS	Excellent comprehensive review of the safety of TCS use with no stone left unturned. Sound methodology, conclusions and acknowledgment of limitations. Very relevant information for the clinician and prescriber. Great job. It was noted at multiple sites, such as page 10, line 56 and page 11, line 39, page 12, line 55, the message: (Error! Reference source not found.) Not clear if these are supposed to be linked references but this format did not work for the PDF provided for review.
--

VERSION 1 – AUTHOR RESPONSE

Reviewer: 1

Comment

This is a well-reported well-conducted umbrella review for the safety of topical

corticosteroids in the treatment of atopic eczema, which is a very important topic for both patients and clinicians.

Response

Thank you for this comment.

Comment

Consider updating the search as it will have been more than 2 years since the search date by the time of publication.

Response

We have updated the search to 2nd March 2021. On the advice of our Information Specialist, we have used our existing search strategy but focussed the search on Epistemonikos rather than replicating the search in other databases. Epistemonikos is now well-established as a comprehensive database of reviews that regularly searches ten major databases including the Cochrane Library, PubMed and Embase1 thus making the need to search these individual databases redundant.

The search update identified two additional systematic reviews for inclusion, both rated critically low quality according to AMSTAR-2 and these have been added to Appendix 5. From these reviews, we identified and included safety data from 3 additional RCTs but the majority of the studies included in these reviews had already been incorporated into the overview. We were able to obtain additional information about some already included studies such as study duration or age and severity of participants, and this has been added to Appendix 6. Additionally, it became clear that one study described as an observational study in the previously included review should be an RCT. Overall this has resulted in an increase of four RCTs and a decrease of one observational study. These additional studies have not added any safety data on specific adverse events, nor have they changed the main messages arising from the overview. The manuscript, relevant appendices and the study flow diagram have been updated accordingly.

1. Rada, G., Pérez, D., Araya-Quintanilla, F. et al. Epistemonikos: a comprehensive database of systematic reviews for health decision-making. *BMC Med Res Methodol* 20, 286 (2020). <https://doi.org/10.1186/s12874-020-01157-x>

Comment

Consider adding whether authors on this review were also authors in the SRs under consideration in your competing interest section.

Response

Good point. We have added to the competing interest section that two of our authors are also authors on included reviews.

Comment

It may be worth adding the detail over participants and study design in your PICOS study question definition in the introduction.

Response

Agreed. We have amended the aims sentence on page 5 of the manuscript to the

following:

"This umbrella review aims to assess safety (local and systemic adverse events) of TCS compared to other topical treatments, placebo or no comparator in people of any age and gender with atopic eczema".

Comment

The section on comparator/control from the protocol doesn't seem to have been included in the main manuscript.

Response

We apologise for omitting this key information. This is now included in the manuscript on page 7:

"Our intervention of interest was any TCS of any preparation and potency used to treat atopic eczema. For RCTs, the comparisons of interest were any other TCS, the same TCS used in a different way, another topical anti-inflammatory treatment, vehicle, no treatment or a combination of any of these. Comparisons with non-topical treatments were excluded as we were interested in clinical practice decisions regarding alternatives to TCS."

Comment

It would be really useful if you could summarise exactly how many studies, and participants, had longer term, say >3/12, data available for safety. Further to this point, it may be worth presenting or framing AEs as short term AEs vs long term AEs. Given the majority of the evidence is in the short-term, the findings could be more reassuring for short-term AEs, say folliculitis, infections, and not so informative for skin thinning, telangiectasia and systemic AEs.

Response

Thank you for this helpful suggestion and we have added the following to the results section. A total of 14 RCTs (5,874 participants) and 5 cohort/observational studies (4,438,698 participants) out of a total of 106 studies included follow up of more than 3 months. This information has been added to the results section.

"A total of 14 RCTs (5,874 participants) and 5 cohort/observational studies (4,438,698 participants) out of a total of 106 studies included follow up of more than 3 months."

Our planned outcomes were cutaneous (immediate and other) and systemic adverse events, and where possible we have presented the results in these groupings throughout the manuscript where they are available. We have maintained the same order throughout i.e. skin thinning, followed by other cutaneous adverse events then systemic adverse events for ease of reading. For each comparison, where available, we have stated the duration of TCS use throughout the results section and in Appendix 6. There is also a paragraph in the discussion section on page 16 discussing the short-term nature of a significant proportion of the safety data.

Although we could include sub-headers (cutaneous immediate, cutaneous other and systemic) in the results section for each comparison, we are concerned this might get repetitive particularly where there are no data available under the header, but we are happy to do this if the editor would like this included. To make it clearer, we have revised the outcomes section of the methods to emphasise the outcomes of interest.

Comment

How was a "systematic" review defined? (in protocol but not in manuscript – I think it should be included.)

Response

Agreed. Most of the definition from the protocol was included in the manuscript under "Eligibility Criteria" but we have added two additional points; the requirement for clinical outcomes in the review and that reviews of any types of clinical study design were included.

"We included systematic reviews that presented data on the safety of TCS used to treat people of any age and gender with atopic eczema, had clinical outcomes, searched at least one database and provided a reproducible search strategy. Systematic reviews of any types of clinical study design were included."

Comment

How were skin thinning and steroid withdrawal defined in the studies and in the SRs?

Response

As this was an overview of reviews, we are only able comment on what was reported in the reviews. Any description of how skin thinning was measured or defined if given in the review is included in Appendix 6. We would have taken the same approach with steroid withdrawal if we had found any relevant studies to include.

Comment

Did the authors consider formulation, e.g. foam vs ointment vs cream, for subgroup analysis? It could be relevant for short-term AEs, e.g. sensations of skin burning. If not, is this a limitation to consider?

Response

We reported the analyses that had been conducted as part of the original reviews as this was an overview of reviews. We did not specify in our protocol that we would look for subgroup analyses based on preparation of the TCS, although we can see that this could be a relevant subgroup analysis for adverse event data. We have added the pre-specified subgroup analyses to the methods and described lack of available data on these as a limitation in the discussion. Detail on preparation used in each study is provided in Appendix 6 where this was available from the reviews.

Comment

There are reference errors on line 56, page 8; line 39, page 9; line 54, page 10.

Response

Apologies and thank you for pointing this out – we think this error must have happened during the conversion to PDF and these have now been corrected.

Comment

What were the methods used for quality assessment? Were they also done in duplicate?

Response

We conducted quality assessment of the included reviews using the AMSTAR-2 tool, and this is described in the methods section under "Assessment of quality of included systematic reviews". We have added in the sentence to explain this was carried out in duplicate.

Comment

Any assessment of meta-biases across studies?

Response

In our methods we have stated "Data on the quality of individual studies (e.g. risk of bias) and the quality of evidence (e.g. GRADE16) were also extracted where presented in the review, but undertaking these quality assessments for individual studies was not within the remit of this overview." We extracted risk of bias assessments where reported and stated in the Results section "Risk of bias assessments were available from the reviews for 63 RCTs, most reporting at least one domain rated as high or unclear risk". We have now included the common reasons for downgrading on the risk of bias assessments.

"Most of these assessments rated at least one domain as high or unclear risk, most noticeably selection bias from lack of allocation concealment, performance bias due to lack of blinding of participants and detection bias due to lack of blinding of outcome assessors."

Comment

A presentation or discussion of the overlapping of studies between SRs may be useful.

Response

How we dealt with the potential for overlapping data is covered in the methods section under eligibility criteria, but we have amended this slightly to be clearer that this was how we avoided any duplication of data or studies in the overview:

"Multiple reviews on the same topic were included, except for "abridged" versions of the same review where no additional data were reported. To avoid duplication of data, for each comparison, the review that included the highest number of studies on that comparison and therefore appeared the most comprehensive was taken as the primary review and other included reviews were checked for additional studies and data".

Comment

I would suggest maybe caveating the first paragraph with your finding of lack of longer-term data, as you have done in the conclusion, as I am not sure this is really reliable data to reassure patients and aid adherence given the paucity of longer-term data and low quality of SRs.

Response

We agree with your point and have amended the first paragraph of the discussion to address the lack of long-term data to balance the findings with the limitations, which now reads:

"Skin thinning and effects on growth concern many people with eczema and parents of children with eczema when using TCS. However, we found no evidence of skin thinning when TCS were used intermittently "as required" to treat flares or as "weekend therapy"

to prevent flares, although the majority of data was from short-term studies.(5) Similarly, we found no evidence of growth retardation or clinically significant adrenal suppression but the only data available was from one 5 year that included 1213 children using TCS. (62) Other studies only reported biochemical signs of adrenal suppression. Adherence to TCS treatment is known to be poor and these findings, particularly around skin thinning, may encourage appropriate use of TCS and therefore improve treatment effectiveness and patient benefit. (119)

Comment

How long should RCTs include for follow-up at a minimum in your suggestion in the conclusion? Worth being specific here. How about cohort studies?

Response

Thank you for this suggestion as the lack of long-term data is an important finding of this overview. However, we would prefer not to suggest a minimum duration for either RCTs as this would at least partly depend on the research question and other aspects of the trial design. However, as RCTs are not necessarily the ideal research method for assessing long-term safety outcomes it is important that we do also mention cohort studies as you suggested. We have included a statement to encourage researchers to consider a longer duration and added this to the discussion (see below).

Comment

I suggest adding "in the short-term" to your last sentence.

Response

We have done this, but to accommodate the addition of the point above, please note this is now the second sentence in the paragraph.

The final paragraph of the discussion now reads:

"In summary, we found no evidence that TCS cause harm when used intermittently "as required" to treatment eczema flares or as "weekend therapy" to prevent flares and this should support the use of TCS in the management of eczema. We found that the adverse events of greatest concern to patients and clinicians, such as skin thinning, are uncommon with short-term use of TCS. However, high-quality evidence was limited, particularly for long-term use. Rather than follow-up of perhaps just a few weeks, future RCTs should include longer follow-up to enable better safety assessment. However, it should be noted that longer-term prospect observational studies are better placed to explore longer-term safety of TCS and should be designed with years rather than months of follow up to add useful information to the field.

Perhaps equally as important as duration of follow up in trials is resolution of adverse events which is often not reported. For adverse events such as biochemical signs of adrenal suppression, it is crucial to know if the effect is transient and levels return to normal once the TCS is stopped, particularly as it is not clear how to interpret the clinical relevance of these."

Reviewer: 2

Comment

Excellent comprehensive review of the safety of TCS use with no stone left unturned. Sound methodology, conclusions and acknowledgment of limitations. Very relevant information for the clinician and prescriber. Great job.

Response

Thank you very much for this comment, especially as you think it will be so useful clinically.

Comment

It was noted at multiple sites, such as page 10, line 56 and page 11, line 39, page 12, line 55, the message: (Error! Reference source not found.) Not clear if these are supposed to be linked references but this format did not work for the PDF provided for review.

Response

Apologies for this error and thank you for pointing this out – we think this error must have happened during the conversion to PDF and these have now been corrected.

VERSION 2 – REVIEW

REVIEWER	Yiu, Zenas Salford Royal NHS Foundation Trust
REVIEW RETURNED	15-Apr-2021
GENERAL COMMENTS	Thank you for the comprehensive replies and amendments.